# Rotation Invariant Graph Neural Network for 3D Point Clouds

**Alexandru Pop, Victor Domșa and Levente Tamas \***

Automation Department, Technical University of Cluj-Napoca, 400114 Cluj-Napoca, Romania
* Correspondence: levente.tamas@aut.utcluj.ro

**Abstract:** In this paper we propose a novel rotation normalization technique for point cloud processing using an oriented bounding box. We use this method to create a point cloud annotation tool for part segmentation on real camera data. Custom data sets are used to train our network for classification and part segmentation tasks. Successful deployment is completed on an embedded device with limited processing power. A comparison is made with other rotation-invariant features in noisy synthetic datasets. Our method offers more auxiliary information related to the dimension, position, and orientation of the object than previous methods while performing at a similar level.

**Keywords:** computer vision; object part segmentation; classification





## 1. Introduction

The recent increase in availability of depth sensors and 3D model databases such as Shapenet [1] has lead to significant interest in 3D computer vision. Processing on point clouds can be made on the order of individual points, as performed in semantic segmentation, or on the entire set, as performed in classification. A necessary element for neural networks applied to point clouds is the ability to create local and global features. Local features synthesize information from the neighborhoods of each point, whereas global features describe the information from all points, similar to a signature of the input. Local features can improve the robustness of the network to noise. Examples of noise that affect the input point clouds from real cameras can be grouped as Gaussian, orientation, and occlusion noise [2].

Graph neural networks have recently been successfully applied in point cloud applications [3–9]. These networks use graphs built using distance between points, which is rotation invariant. Thus, architectures such as those in [3,8,10] use graph neural networks in combination with rotation invariant features to completely mitigate rotation noise.

Our work is based on the Regularized Graph CNN for Point Cloud Segmentation (RGCNN) network [5]. It uses three spectral graph convolution modules to compute local features which are then used for classification and semantic segmentation. As explained in [5], the RGCNN network displays remarkable resistance to noise and a reduction in the number of necessary parameters compared to state-of-the-art PointNet [11], PointNet++ [12], and Dynamic Graph CNN (DGCNN) [4].

Table 1 shows the inference times of RGCNN and DGCNN networks for classification and part segmentation tasks on an Nvidia AGX Xavier embedded device. The results show that for a lower number of points, RGCNN is considerably faster than DGCNN, while at 2048 points, it performs worse in both scenarios. Since real-time speed is the goal of running machine learning models on embedded devices, and since RGCNN is more resilient to noise, as described in [5], we chose to perform the rest of the experiments with RGCNN as the main model.

**Table 1.** Forward time in milliseconds on an Nvidia AGX Xavier 32 GB embedded device. For a lower number of points in the input point cloud, RGCNN is considerably faster then DGCNN for both classification and part segmentation task. Even though the forward time for point clouds with 2048 points is higher for RGCNN, this number of points might not be suitable for real-time application deployed on embedded devices as the preprocessing time also increases.

| Network | Fw. (ms) 512 Points | Fw. (ms) 1024 Points | Fw. (ms) 2048 Points |
|---|---|---|---|
| DGCNN Cls | 15.82 | 23.28 | **52.71** |
| DGCNN Part Seg | 13.64 | 17.17 | **38.89** |
| RGCNN Cls | **4.91** | **12.09** | 57.39 |
| RGCNN Part Seg | **7.27** | **15.94** | 61.78 |

We combined RGCNN network with a rotation normalization procedure similar to the eigenvector matrix method described in [3,8]. Our method is highly resistant to random rotations without any data augmentation while at the same time being able to improve the resistance to Gaussian noise and occlusion noise through data augmentation training.

## 2. Related Work

### 2.1. Graph Neural Network for 3D Data

Graph neural networks have been used recently in numerous applications [4–7,10,13,14]. The flexible nature of a graph permits its usage from data sets concerning large social networks to smaller networks that describe the chemical bonds of a molecule. Graph neural networks use the correspondences between elements instead of focusing on individual elements. These correspondences help create neighborhoods and local regions, which greatly enhance the predictive accuracy of the resulting features. This adaptability to different types of data and the possibility to create neighborhoods prove useful in analyzing 3D point collections describing shapes called point clouds. By converting point clouds to graphs, these types of neural networks can improve the information obtained from neighborhoods of points.

Classical pointwise networks use multilayer perceptrons (MLP) to obtain features. A prime example of this method is the PointNet network introduced in [11]. It sequentially applies multilayer perceptrons and fully connected layers to obtain the features. An improvement was made in PointNet++ [12] where groupings of points are made before applying PointNet. The features built with MLPs lack information from extended point neighborhoods. Ref. [15] modifies PointNet to learn local features better in order to improve the semantic segmentation task on trees. Ref. [16] improves the segmentation results of neural networks by fusing semantic global features with semantic edge features obtained from an encoder–decoder system. In this way, the resulting point clouds have better boundaries and, as such, the mean intersection over union (mIoU) increases. Graph convolutional networks are a solution to mitigate the lack of information from neighborhoods.

Graph convolutional networks are divided in two main categories: spatial and spectral. Spatial graph convolution networks iterate through each vertex and use neighbors to compute the convolution. They mimic the sliding kernel of traditional convolution networks, but instead of taking neighboring points on a grid, they use the neighbors of each vertex based on the graph. These methods have the advantage of only needing the neighborhood of each vertex for a convolution, not the entire graph. Thus, parts of the graph relevant to the current vertex can be used without loading the entire graph. An example of a spatial-based graph convolution network applied on point clouds is DGCNN [4].

Some recent works such as [17] also study the performance of deeper architectures. They study the problems of vanishing gradients and over-smoothing. These challenges are addressed by borrowing more concepts from classical CNNs such as residual connections and dilated convolutions and graphs. The results show an increase in mIoU for Semantic Segmentation tasks compared with shallow GCN.

Spectral graph convolution networks use the weighted adjacency matrix to compute the graph Laplacian that synthesizes the interactions between the vertices. Using a function that changes the Laplacian changes the interactions between points, leading to different neighborhood creation and clustering. Using a weighted sum of Laplacian powers, it creates features based on combinations of neighborhoods. These new features can help cluster the vertices leading to a better classification or semantic segmentation. Figure 1 shows an example use of spectral graph neural network in point cloud processing.

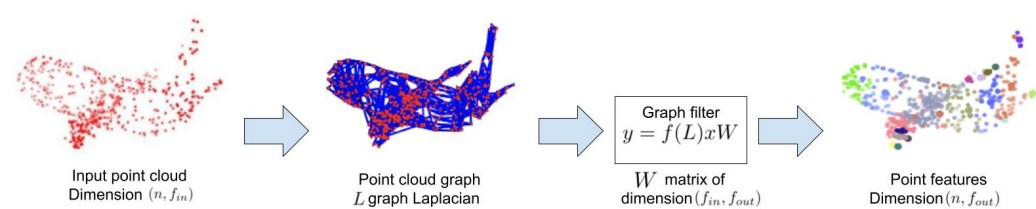

**Figure 1.** Spectral graph convolution for $n$ points with $f_{in} \in \mathbb{N}$ input number of features, given by the size of the previous layer and $f_{out} \in \mathbb{N}$, the desired output feature size.

The downside of spectral graph convolution is that it requires the entire graph to make the convolution because it utilizes the full weighted adjacency matrix. One way to perceive the thematic difference between the two convolution methods is to view the spatial convolution as a local operation applied to each vertex using their neighborhoods. The spectral convolution is a global operation applied directly on all vertices that take into consideration the information from the graph Laplacian.

Examples of spectral graph neural networks applied on point clouds can be seen in [5–7,10,18]. A different spectral graph convolutional network with more representation power is described in [19]. This module has not been used in point cloud applications as opposed to Chebnet [5] and GCN [20].

A more thorough survey on graph neural networks can be found at [14].

### 2.2. Rotation Invariant Classification and Part Segmentation Networks

Rotation invariance is obtained either using a rotation normalization algorithm or by computing rotation invariant features from the point cloud. Rotation normalization implies a computation of a rotation matrix specific to each point cloud that places the input in a canonical position and can be treated as a preprocessing of the input data.

Rotation invariant features are built by using neighborhoods and relationships among points. If any rotation applied on the point cloud does not change the underlying neighborhoods and relationships among points, the entire network displays rotation invariance. A trade-off occurs between the prediction power of the features and the cost of computing them. Examples of handcrafted rotation invariant features are described in [21–26]. These are handcrafted features computed on knn distance neighborhoods of a sub-sample of points from the input. Ref. [27] extends the rotation invariant Point-Pair-Feature (PPF) from [21] such that the local pose information from patches is used in the convolution process. This leads to global rotation invariance and better prediction if only a subset of patches is rotated.

Another example of a rotation invariant feature is the sorted Gram matrix applied in [28]. For a point cloud $P \in \mathbb{R}^{N \times 3}$, it computes the Gram matrix $G = P^\top \cdot P$, $G \in \mathbb{R}^{N \times N}$. Sorting ascending each line of the Gram matrix endows it with permutation invariance. The sorted Gram matrix contains the $N \times N$ relationships among all points and these relationships are rotation invariant. One clear vulnerability of this method is that it leads to an increase in the network size depending on the number of points of the input.

Refs. [10,18] use spectral graph convolutional neural networks to create features that are rotation invariant. Ref. [10] uses Chebnet spectral graph convolution modules [29], whereas [18] uses GCN modules described in [20].

Ref. [30] uses two networks to compute local and global rotation invariant features. For both global and local features, they use an SVD decomposition of the point cloud to determine the best orientation and using a neighbor point resolves the ambiguity of the orientation. The global rotation invariance is similar to the PCA rotation normalization. Using the furthest point from the mean as an anchor point, they achieve the pose disambiguation by comparing the axes in the rotated point cloud with the vector from the center to the anchor point.

Our proposed solution has a similar technique as [3,8], which perform a rotation normalization. It uses PCA to compute a bounding box that best fits the dimensions of the point cloud. The orientations of the length, width, and height of the bounding box give the rotation relative to the rotation matrix of the box relative to the global frame. This method offers an intuitive visualization for the rotation normalization as the rotation neutralizes the inclination of a bounding box. The same object with a random rotation has the bounding box oriented slightly differently. Knowing the inclination of the bounding box, we can multiply with its transpose to align to the coordinate axis.

The papers that use a method similar to us are [3,8,31]. They all use PCA to place a point cloud in a canonical position. For each point cloud $P \in \mathbb{R}^{N \times 3}$, an eigenvalue decomposition is conducted on a matrix $R = P^{\top} \cdot P$, $R \in \mathbb{R}^{3 \times 3}$. By ordering the eigenvalues in descending order and placing the corresponding eigenvectors, the rotation normalizing matrix $U \in \mathbb{R}^{3 \times 3}$ is obtained. Multiplying the input point cloud with its corresponding rotation normalization matrix $P'$ places it in a canonical orientation regardless of other rotations applied on the point cloud.

A paper of interest is [32], which computes features that display rotation invariance. The features are computed in the neighborhoods of sampled points from the point clouds. PCA algorithm is used on each neighborhood to determine a canonical position. They solved the problem of ambiguity of PCA in each neighborhood by choosing a predefined anchor point.

As described in [3], PCA has inherent ambiguities in the order of the eigenvectors and the sign. Ref. [8] creates four views of the point cloud by using the PCA rotation matrix and iteratively changing the sign of one eigenvector. They apply a shared DGCNN [4] network on each view and aggregate the information using a self-attention module and average pooling. We tested a similar architecture for classification, but we used the RGCNN module [5]. The big downside of using four views is that the network needs much more memory than one view. For embedded devices with limited memory, this option can become problematic. Li et al. [3] also switched the order of the eigenvectors leading to 24 different views. They used a simpler 1D convolution network to aggregate the information from each of the 24 rotated point clouds to eliminate ambiguity. Ref. [31] created 120 poses from the 4 obtained by changing the sign of the eigenvectors. A pose selector module chooses the best pose used in the rest of the network. For local features, it uses hand-crafted rotation invariant features obtained in knn neighborhoods, similar to [33].

Unlike the PCA rotation normalization [3,8] methods described above, our method offers additional information that describes the object. The oriented bounding box has an approximate width, length, and height of the shape sampled by the point cloud, along with a position and an approximate orientation given by the orientation of the bounding box. Using the oriented bounding box method, a robot has a first-order approximation of the volume of an object along with its orientation. In PCA rotation normalization, only the orientation is used. A visual comparison between the Bounding Box normalized rotation and the Eigenvector normalized rotation is shown in Figure 2. The original point cloud is rotated in different positions. Both methods obtain a canonical rotation but the oriented bounding box also offers volumetric information, which can be used in robotic settings. The points obtained from the convex hull of the point cloud are used for the oriented bounding box computation, which leads to an approximation of the minimum volume bounding box.

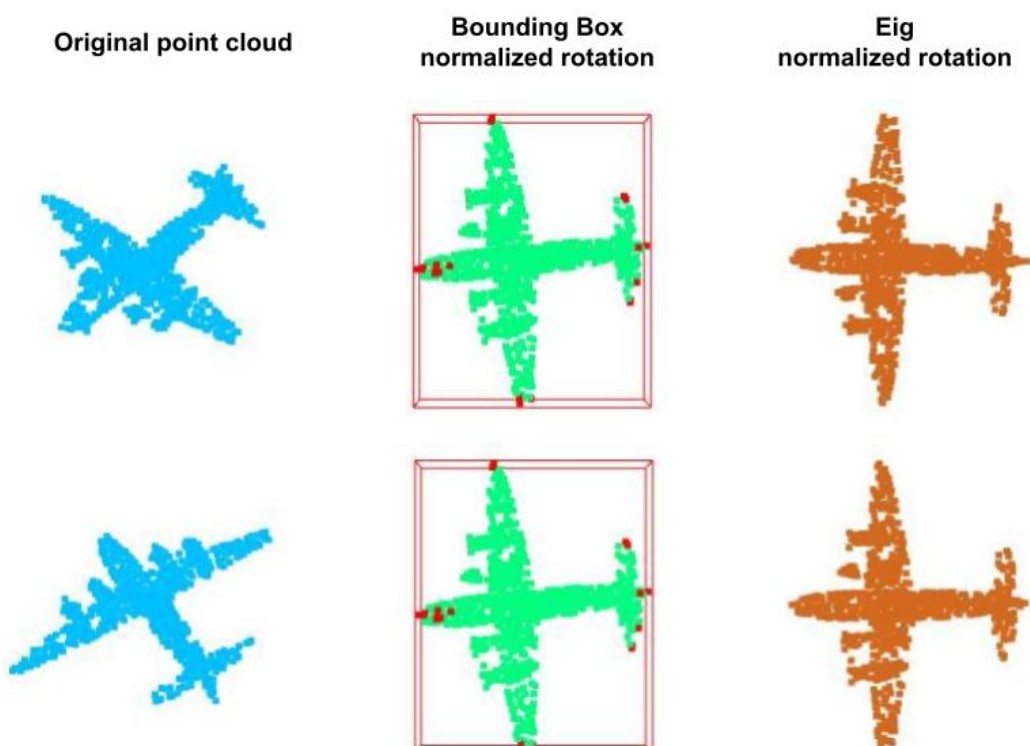

**Figure 2.** Visual comparison between Bounding Box and PCA preprocessing.

### 3. Theoretical Background

*3.1. Background for Graph Neural Networks*

A graph $G$ is described as a collection of vertices $V$ with $n$ elements having correspondences. The existence of a relationship between two vertices is an edge of the set $E$. The corresponding values for each relationship between vertices are given by the weighted adjacency matrix $A \in \mathbb{R}^{n \times n}$, thus $G = (V, E, A)$. To create a graph, the set of vertices must be established, then determining which vertices have a relationship, and finally computing the weighted adjacency matrix for the entire graph. A typical method for computing the adjacency matrix $A$ is found in (1).

$$A_{i,j} = e^{-\frac{\|p_i - p_j\|^2}{\sigma^2}} \tag{1}$$

where $\|p_i - p_j\|^2$ represents the Euclidean distance, while $\sigma > 0$ is given in [6,34] as a variable parameter. The degree matrix $D$ can be computed from the weighted adjacency matrix by summing the elements on each line and creating from it the diagonal matrix. The first iteration of the Laplacian $L$ is $L = D - A$. To normalize the Laplacian graph, we divide by the degree matrix as $L_{norm} = D^{-1}(D - A)$. To ensure that the Laplacian graph matrix is symmetric, the division by $D$ is carried out according to (2).

$$L_{symm} = D^{-1/2}(D - A)D^{-1/2} \tag{2}$$

Graph convolution can be seen as an extension of classical convolutional networks to graph data. Traditional convolutional networks use inputs in a fixed grid structure, whereas graph data do not have a spatially localized structure and must rely on the relationships. All traditional convolutional networks require a smaller fixed-size filtering kernel that is moved sequentially through the grid structure. In graph neural networks, vertices that have a relationship are considered neighbors and the number of neighbors of one vertex can vary greatly in number. Depending on the graph, some vertices can have many more neighbors than others.

### 3.2. Spectral Graph Convolution Neural Network

Spectral graph convolution neural networks in point cloud applications use the modules designed by [29,35]. The spectral convolution module described by [35] decomposes the Laplacian into eigenvalues and the corresponding eigenvectors as $L = U\Lambda U^{\top}$, where $U \in \mathbb{R}^{n \times n}$ is the matrix of eigenvectors and $\Lambda \in \mathbb{R}^{n \times n}$ is the diagonal matrix of eigenvalues. An input signal is considered to be $X \in \mathbb{R}^{N \times f_{in}}$, where $f_{in} \in \mathbb{N}$ is the number of features for each point. $F(X) = U^{\top}X$, $F(X) \in \mathbb{R}^{n \times f_{in}}$ is equivalent to converting a signal $X$ from the spatial domain to the spectral. Taking into account the output signal $\hat{X} \in \mathbb{R}^{N \times f_{in}}$ $F^{-1}(\hat{X}) = U\hat{X}$ gives us the inverse operation. By converting an input in the spectral domain, we can use a filter $g$ to manipulate the values in the spectral domain and then use an inverse operation to convert it back into the spatial domain. Thus, the first interpretation of the spectral convolution described in [35] is found in (3).

$$\hat{X} = F^{-1}(F(X) * F(g)) \tag{3}$$

"$*$" is the Hadamard product and $g \in \mathbb{R}^{n \times f_{in}}$ is a filter of the same dimension as the input. The simplification is carried out considering that the spectral filter $g' \in \mathbb{R}^n$ is a 1-dimensional vector applied to each of the $N$ channels of the input. In this way, the filter $g' \in \mathbb{R}^n$ and the input vector $X \in \mathbb{R}^{n \times F_{in}}$ are of different dimensions. The spectral convolution is rewritten in (4).

$$\hat{X} = U diag(Ug')U^{\top}X \tag{4}$$

$Ug' = \theta \in \mathbb{R}^n$ is the parameter of the network which must be fine-tuned and since the eigenvectors depend on the corresponding eigenvalues, the spectral filter can be seen as a function of the eigenvalue diagonal matrix $\Lambda = diag([\lambda_1, \lambda_2, \ldots, \lambda_N]^{\top})$, $\lambda_k \in \mathbb{R}$ being an eigenvalue. Thus, the formulation used in graph spectral convolutional networks is described in (5).

$$\hat{X} = Uh_{\theta}(\Lambda)U^{\top}X \tag{5}$$

where $h_{\theta}(\Lambda) = diag(\theta) = diag(Ug')$. By manipulating the eigenvalues, the resulting matrix has graph filtering properties similar to a Fourier decomposition of a signal into multiple harmonic frequencies.

A further improvement is the Chebnet spectral graph convolution module described in [29]. This method avoids the expensive Eigen decomposition by expressing the graph filter as a weighted sum of powers of Laplacian. The spectral graph convolutional module used in the network is based on (6).

$$h_{\theta}(\Lambda) \approx \sum_{k=0}^{K} \theta'_k T_k(\Lambda) \tag{6}$$

where $K \in \mathbb{N}$ is the filter size, $\theta'_k \in \mathbb{R}$ are the filter coefficients, and $T_k(\Lambda) \in \mathbb{R}^{n \times n}$ represents the Chebyshev polynomial applied to the diagonal matrix of eigenvalues. In this way, a function of the eigenvalues is represented as a weighted sum of Chebyshev polynomials of the diagonal matrix of eigenvalues. The Chebyshev polynomials can be computed recursively with the starting terms $T_0(X) = I$, $T_1(X) = X$ and $T_k(X) = 2T_{k-1}(X) - T_{k-2}(X)$. The spectral convolution function is rewritten in (7).

$$h(L) = U\left(\sum_{k=0}^{K} \theta'_k T_k(\Lambda)\right)U^{\top} = \sum_{k=0}^{K} \theta'_k T_k(L) \tag{7}$$

If the output signal $\hat{X}$ has a different number of channels than the input signal $X$, a further linear model $W \in \mathbb{R}^{f_{in} \times f_{out}}$ is needed to transform the input signal $X \in \mathbb{R}^{n \times f_{in}}$ to the output signal $\hat{X} \in \mathbb{R}^{n \times f_{out}}$, where $f_{out} \in \mathbb{N}$ is the number of features for each point

in the output vector. Thus, applying also the sigmoid function, the spectral convolution module is described in (8).

$$\hat{X} = \sigma\left(\left(\sum_{k=0}^{K} \theta_k T_k(L)\right) XW + b\right) \tag{8}$$

$b \in \mathbb{R}^{n \times f_{out}}$ is the bias vector. A visualization of the process can be seen in Figure 1.

Ref. [5] uses a different linear model $W_k$ and corresponding bias $b_k$ for each power of the Laplacian instead of a single one $W$ and $b$ applied on the sum. This leads to the expression described in (9)

$$\hat{X} = \sigma\left(\sum_{k=0}^{K} T_k(L) X (W_k + b_k)\right) \tag{9}$$

where $W_k \in \mathbb{R}^{f_{in} \times f_{out}}$ and $b_k \in \mathbb{R}^{f_{in} \times f_{out}}$. We can observe that $\sum_{k=0}^{K} \theta_k T_k(L) = \alpha_K L^K + \alpha_{K-1} L^{K-1} + \cdots + \alpha_1 L + \alpha_0 I$, where $\alpha_i \in \mathbb{R}$ are coefficients corresponding to each Laplacian power graph. Therefore, with this method, any graph spectral filter is written as a weighted sum of $K$ powers of the graph Laplacian $L$. Applying the Laplacian at power one means taking into consideration a 1-hop neighborhood, the second power bringing the effects of a 2-hop neighborhood, and so on until a chosen value $K$. This offers the possibility of computing local features and taking into consideration the relationships between points, controlling the size of the required neighborhoods.

## 4. Methodology

*Proposed Method*

Our network uses a rotation normalization algorithm that computes the bounding box with the lowest volume that best fits the point cloud. The oriented bounding box is computed using the PCA of the convex hull. A convex hull is a mesh that envelops all the points of the point cloud. The mesh has as vertices a subset of the point cloud. An example of a convex hull for the point cloud is seen in Figure 3a and the resulting vertices are seen in Figure 3b.

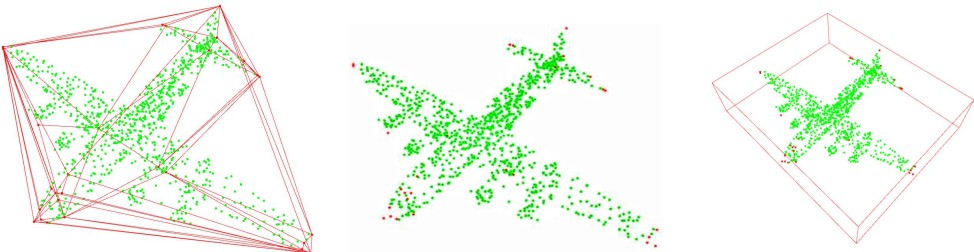

(**a**) Convex hull creation　　　　　(**b**) Selected points　　　　　(**c**) Oriented Bounding Box

**Figure 3.** Oriented bounding box computation. In (**a**), the convex hull is created. In (**b**), the points of the hull are selected and used in a PCA algorithm. The resulting bounding box is shown in (**c**).

Once the vertices of the mesh have been established, PCA analysis is applied on these points, leading to the eigenvalues and corresponding eigenvectors ordered in descending order. The bounding box is obtained by bringing the mesh in a canonical rotation and then finding the maximum and minimum values on the $X$, $Y$, and $Z$ axes. The canonical rotation is obtained by multiplying the convex hull vertices with the transpose of the eigenvector matrix. Afterward, the dimensions and corner points are computed from the minimum and maximum values on each axis. The bounding box orientation is given by the eigenvector matrix. An example of the oriented bounding box is given in Figure 3c.

The convex hull of the point cloud is computed using Open3D, which bases its method on the implementation from Qhull. For each point cloud, we obtain the oriented bounding box and multiply it with the inverse of the rotation matrix, obtaining a rotation

normalization of the input in a canonical position. We use the rotation normalized point cloud with the RGCNN network proposed in [5] for classification and part segmentation, as seen in Figure 4.

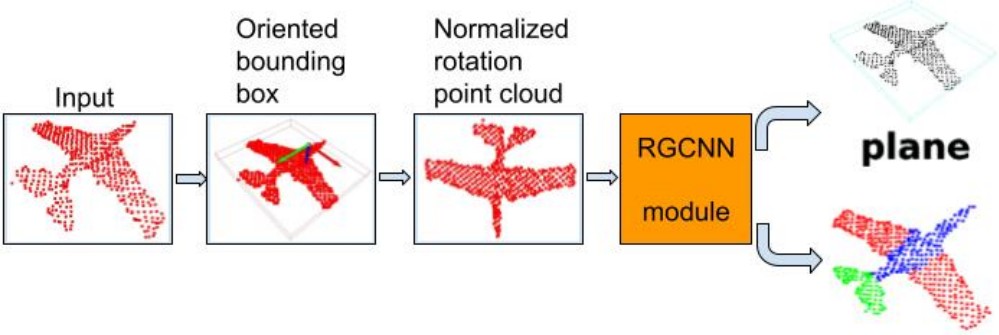

**Figure 4.** Proposed method with classification and part segmentation.

## 5. Datasets

### 5.1. Synthetic Datasets

To test our method for the synthetic data, we used the Modelnet40 dataset to train and test the classification model and Shapenet for the part segmentation model. We tested the networks on three categories of noise: Gaussian, rotation, and occlusion noise. Because the oriented bounding box needs 3D objects, we eliminated four classes of objects that contain flat point clouds, such as curtain, door, person, and plant. For each task, the first test was with models trained on ground truth data and tested on data affected by noise. The second test was with models trained on data augmented with noisy datasets and tested on data without noise.

#### 5.1.1. Gaussian Noise

We used Gaussian noise with zero mean and changed the standard deviation $\sigma$. We created a new Modelnet40 dataset with noise added according to each change in $\sigma$. The selected values were $\sigma \rightarrow [0.02, 0.05, 0.08, 0.1]$. Examples of point clouds with Gaussian error are in Figure 5.

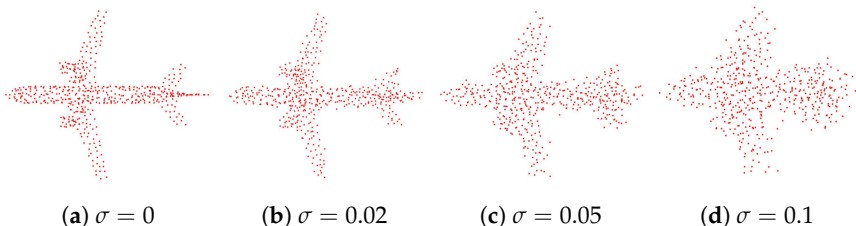

(**a**) $\sigma = 0$         (**b**) $\sigma = 0.02$         (**c**) $\sigma = 0.05$         (**d**) $\sigma = 0.1$

**Figure 5.** Examples of random noise applied on input.

#### 5.1.2. Orientation Noise

We separately added random rotations on the $X$, $Y$, and $Z$ axes to simulate the rotation errors from a real camera. All point clouds were rotated randomly on the $X$ axis followed by the $Y$ and $Z$ axes. All rotations were according to a chosen maximum angle, each selecting a number between $[-angle, +angle]$. The angles chosen for which a separate data set was created were: $angle \rightarrow [10, 20, 30, 40]$. An example point cloud with random rotations is shown in Figure 6.

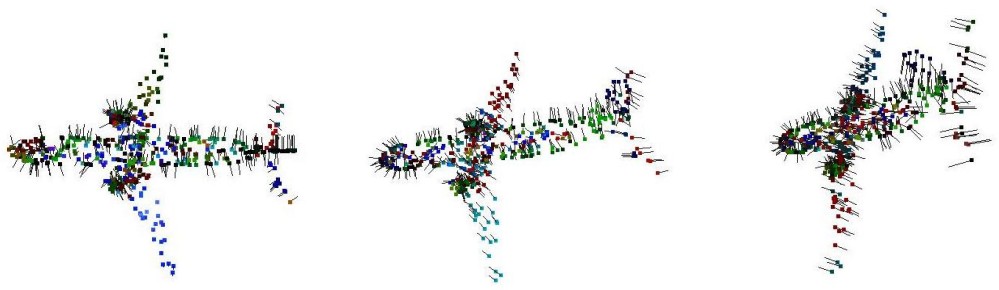

(**a**) 0 degree rotations　　　　(**b**) Random 20 degree rotations　　　(**c**) Random 30 degree rotations

**Figure 6.** Examples of random rotations applied on input.

### 5.1.3. Occlusion Noise

Occlusion noise affects real 3D cameras. Obstructing objects or noise can lead to parts of the object point cloud being missing, similar to holes in the shape. For classification and part segmentation settings, we suppose that we can obtain the points corresponding to the object, but parts of the shape are missing.

We simulate occlusion noise by randomly choosing a point from each point cloud and removing the neighboring points in a ball-radius neighborhood. Increasing the radius leads to a larger neighborhood around the chosen point and consequently to more points being removed. Before removing the points, we need to ensure we have enough remaining points to apply furthest point sampling to obtain the fixed sized point cloud necessary for the input. To do this, we first sample 3000 points from each mesh in the Modelnet40 and Shapenet databases. We remove points according to the occlusion assumption described above and finally use furthest point sampling on the remaining points in order to reach the fixed sized point clouds. A selection of point clouds with occlusion noise can be seen in Figure 7.

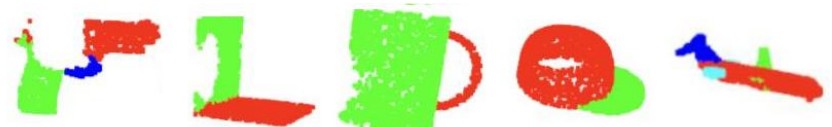

**Figure 7.** Examples of occlusion noise on Shapenet [1].

### 5.2. Custom Camera Datasets

We used a Time of Flight camera to gather point clouds from a scene. The scene consisted only of the object and the floor, which allows us to segment out the object point cloud by removing the floor points approximated by the largest plane. Moving the camera around the object, point clouds from different orientations representing the same object class are obtained.

Segmentation of the floor plane can lead to a variable number of points in the object point cloud. If the object has more points, we do a furthest point sampling to reach the required number of points. If the point cloud has less points, we compute the normals from the object points and create a mesh from which we sample the required number of points. Another approach to obtain the missing points much faster is to randomly sample points from the available ones and add them to the object points. Effectively, we randomly double some points in order to reach the required number of points.

### 5.2.1. Classification Custom Dataset

The point clouds in the dataset were chosen from five classes of objects, namely: chair, can, bag, headset, shoe. There were 400 points obtained from the segmentation of the object. We sampled 256 points using furthest point sampling. Examples of point clouds from each category can be seen in Figure 8.

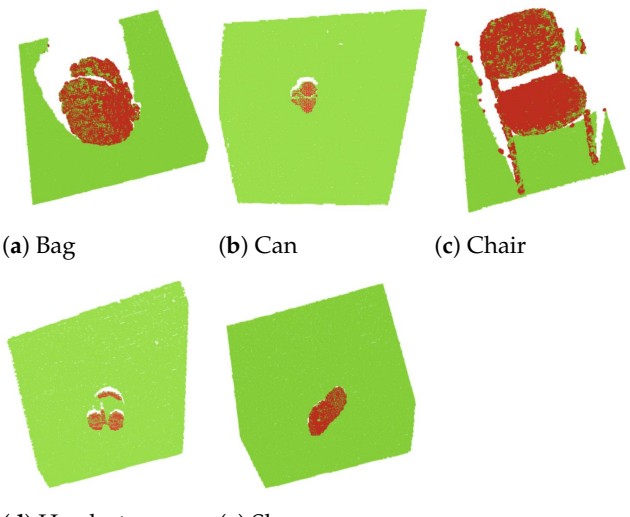

(**a**) Bag       (**b**) Can       (**c**) Chair

(**d**) Headset       (**e**) Shoe

**Figure 8.** Real camera classification dataset.

5.2.2. Part Segmentation Custom Dataset

Similar to the real camera classification dataset, we isolated the points from an object by removing the floor points using RANSAC.

To annotate the points, we selected a point cloud in the canonical pose that we called **target**. For the rest of the point clouds in the object category, we brought them to the canonical pose and used the point cloud distance function from Open3D to measure the lowest distances from the point cloud to the target, resulting in a vector of distances **dist** $\in \mathbb{R}^N$.

Then, we rotated the point cloud 180 degrees on the *X* axis, the *Y* axis, and the *Z* axis, and used the target for each distance function, leading to the distance vectors **distX**, **distY**, **distZ**. We concatenated the distances that lead to a vector **dist**$_{\textbf{all}} \in \mathbb{R}^{N \times 4}$. For each row, we compute the index of the smallest number and then count for each index the number of times that it was selected as the smallest. The justification is that since the target model is in the same object class as the input point cloud, if the target and the input have the same orientation, then the distances will be the lowest. A visualization of the entire process can be seen in Figure 9. After the point clouds are rotated, we used a series of box-shaped pass-through filters with customizable orientation to select parts of the point cloud we are interested in. These parts are labeled depending on how the pass-through filters are configured. An example of sequential labeling using custom pass-through filters can be seen in Figure 10.

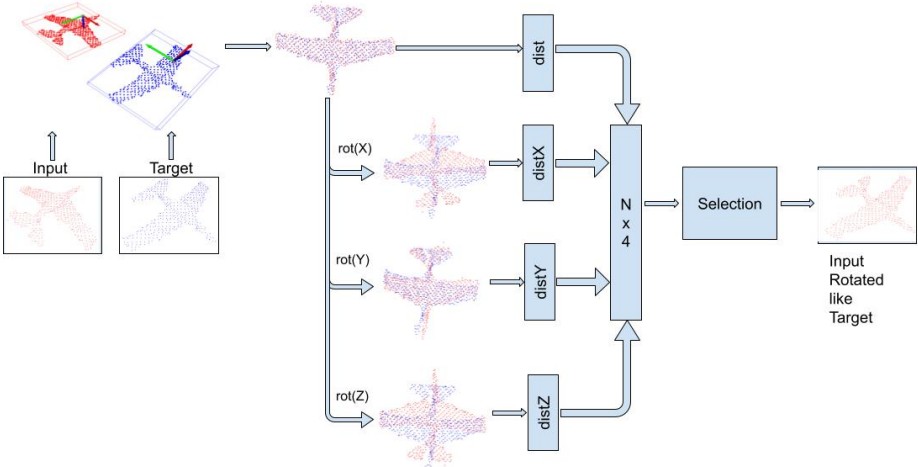

**Figure 9.** Rotation placement in part segmentation.

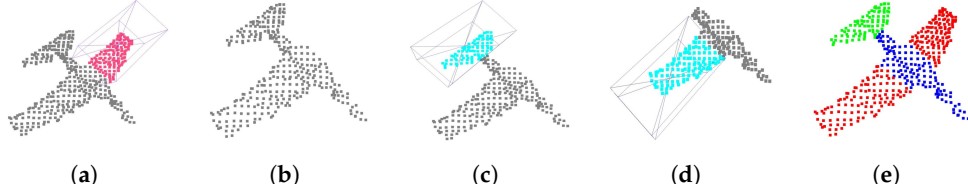

|  (**a**)  |  (**b**)  |  (**c**)  |  (**d**)  |  (**e**)  |

**Figure 10.** Sequentially labeling points for a point cloud. (**a**) First pass through; (**b**) remaining points; (**c**) second pass through; (**d**) third pass through; (**e**) final labeled point cloud.

### 5.3. Implementation on Embedded Device Using Real Data

Our setup involves using ROS as the backbone of the application, while the segmentation and classification parts have been written in PyTorch. Part of the data augmentation, including normals computation, was completed using Open3D. The same workflow is used for both segmentation and classification tasks. Our tests are based on the assumption that the object of interest is placed on the largest flat surface captured by the camera. This surface was often the floor. In order for the floor to be removed, we first employ a voxelization downsample algorithm to obtain the largest plane, which can then be removed by using a passthrough filter. The remaining point cloud represents only the objects on the plane. Currently, our method performs classification and segmentation on only one point cloud; therefore, another assumption is required: only one object of interest is captured by the camera at any time. Next, we sample the point cloud in order to obtain a cloud with the same size as required by the model. In the case of downsampling, we can perfom a Furthest Point Sampling operation. In the case of the Jetson Nano, which has limited processing power, we chose a random sampling of the point cloud, as it is much faster. For the upsampling procedure, we tested two methods: estimating meshes from the point cloud rather than using Furthest Point Sampling or randomly doubling points from the original point cloud. Again, due to the limited resources of the Nano, we chose the random choosing method. After sampling, we center the point cloud and perform the rotation normalization procedure. We can then estimate the normals on this sampled point cloud and then feed it into the model.

In Figure 11, we show the results of real-time semantic segmentation of point clouds recorded by a depth camera. The dataset was recorded using the same camera and annotated using the bounding box method.

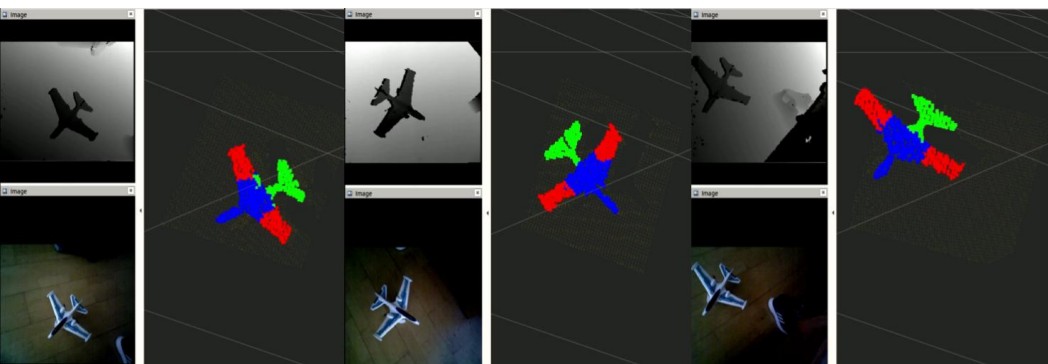

**Figure 11.** Test result for segmentation on Jetson Nano in real time.

### 6. Results

For our tests we used the RGCNN network. The input consists of the point coordinates and the point normals for each point cloud. The RGB data were omitted to analyze only the spatial information. Furthermore, depth images can be recorded in a wider variety of scenarios such as dark or very poorly illuminated places. Lastly, all the base implementations of the other networks ([4,5,11]) use only the spatial information.

### 6.1. Classification

We compared the original RGCNN network with our own extensions, which achieve robustness to rotation noise. The methods can be seen in Figure 12. The tests were conducted on the Modelnet40 dataset. We selected 512 as the number of points for the input point clouds of the networks. We used only the point coordinates in our experiments. At this number it has comparable performance with the networks trained on 1024 and 2048 points while taking half the time of the 1024 point network. The original RGCNN network is shown in Figure 12a and uses $N = 512$ points as input and outputs the selection vector for the class. The RGCNN with an oriented bounding box, illustrated in Figure 12b, contains the preprocessing part, which places a point cloud in the canonical rotation. The pre-processing can also store the calculated bounding box width, length, and height. RGCNN with Gram matrix preprocessing is shown in Figure 12c. Pre-processing converts the input point cloud into a set of characteristics of dimension $N \times N$. This set of features is passed to the RGCNN network, thus needing a change in the first spectral convolution module. For the original RGCNN module, the input feature size for the first spectral convolution is $f_{in} = 3$ and the output feature dimension is $f_{out} = 128$. For the Gram matrix preprocessing, the input size will be $f_{in} = N$ and $f_{out} = 128$. For the multi-view networks shown in Figure 12d,e, the original point cloud is placed in the canonical rotations using the oriented bounding box in Figure 12d and eigenvector decomposition in Figure 12e. Since there is an ambiguity in the selection of the eigenvectors for the second and third dimensions, four possible canonical rotations are considered as detailed in [8]. This means that the input point cloud is placed in a canonical rotation, the distance graph is computed, and the point cloud in the canonical rotation is rotated three times. Thus, the pre-processing and extra rotations will convert the input point cloud of dimension into a batch of four point clouds. The net effect is similar to a four-fold increase in the batch size, but it reduces the effect of the sign ambiguity in the eigenvector decomposition.

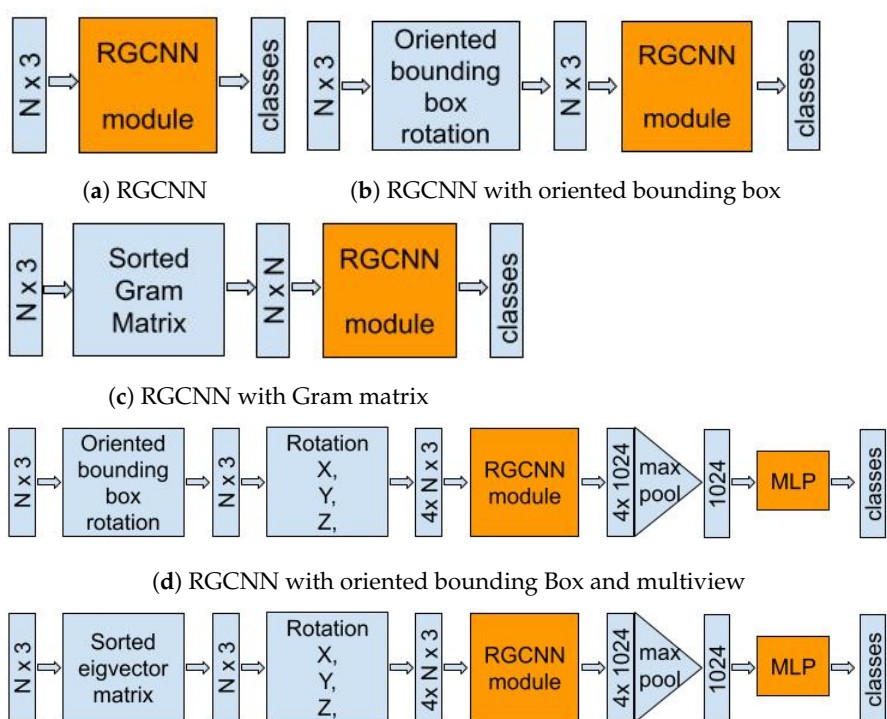

**Figure 12.** RGCNN versions tested for classification.

### 6.1.1. Synthetic Dataset without Noise

To test the robustness to noise of the models, we trained each network on the dataset without noise, and afterward, we tested them on the datasets with Gaussian, orientation, and occlusion noise. The results of the tests can be seen in Figure 13.

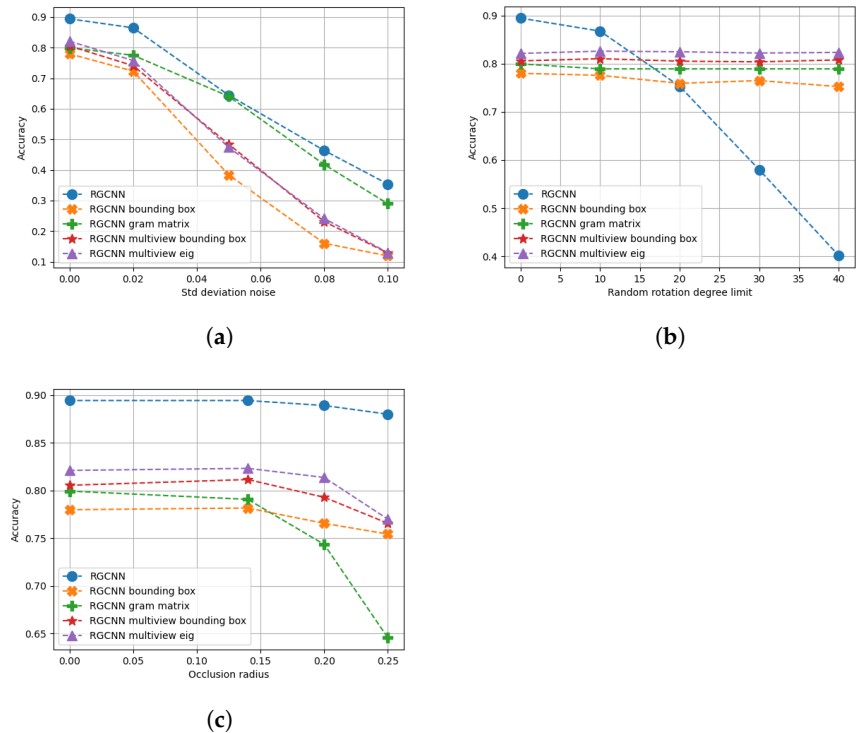

**Figure 13.** Results for classification networks trained without noise. (**a**) Tests for different Gaussian position noise levels. (**b**) Tests for different rotation noise levels. (**c**) Tests for different occlusion noise levels.

### 6.1.2. Synthetic Dataset Trained with Noise

For these tests, we train the networks using the datasets containing noisy data. We train with augmented data according to the noise we want to mitigate. Thus, three training sessions were conducted, the first using position noise, the second using orientation noise, and the third using occlusion noise. The measured accuracy in each test is the overall mean accuracy.

These networks, trained with noisy data, were tested to see if the respective noise was mitigated. The results of these tests can be seen in Figure 14. We observe that the networks trained with position noise had a much higher tolerance to noise. There was no huge accuracy loss for the networks trained with noisy data. The original RGCNN maintained the highest accuracy.

### 6.2. Part Segmentation

Part segmentation tasks are more susceptible to position, rotation, and occlusion noise than the classification tasks. Therefore, extensive tests were performed on RGCNN and the extensions that make the model rotation invariant. The first considered method consists of using the oriented bounding box to normalize rotations, while the second rotates all the point clouds according to its eigenvectors. Since these kinds of noises often appear at the same time in real-life scenarios, it is important to also analyze the performance of said models on datasets augmented with combined noise types.

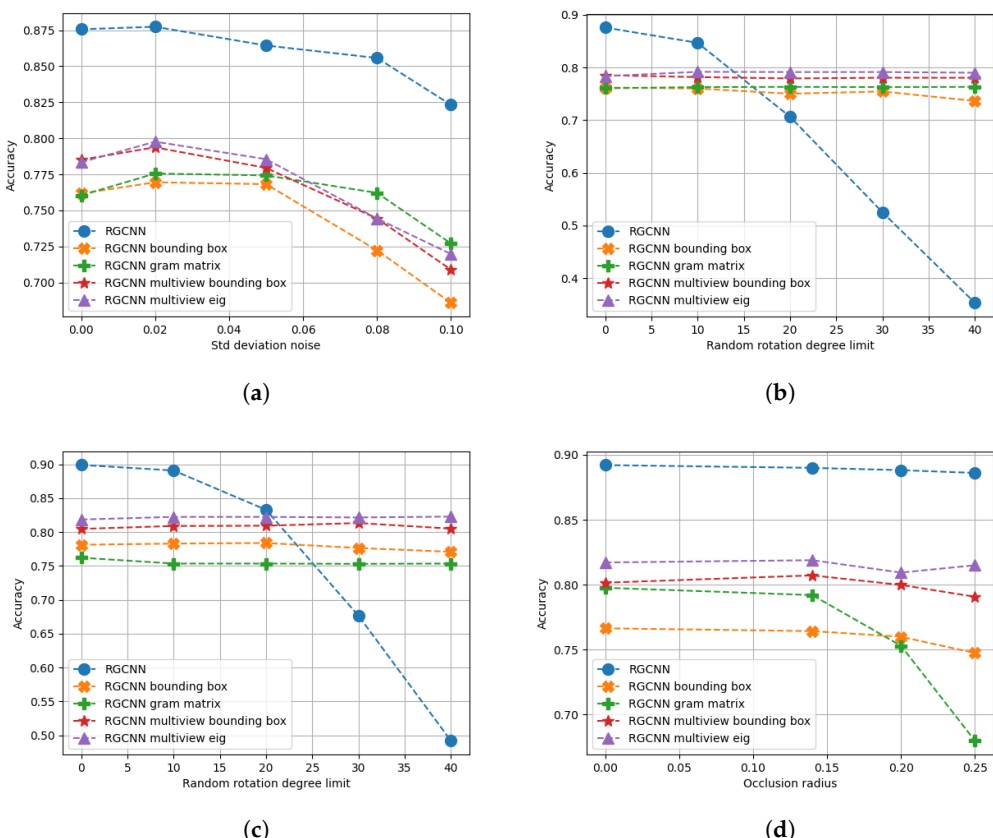

**Figure 14.** Results for classification networks trained with noise. (**a**) Position noise results for models trained with position noise. (**b**) Rotation noise results for models trained with position noise. (**c**) Rotation noise results for models trained with rotation noise. (**d**) Occlusion noise results for models trained with occlusion noise.

6.2.1. Synthetic Dataset without Noise

Firstly we analyzed the case with models trained on the original, unaltered dataset and tested the model on the dataset augmented with Gaussian, rotation, and occlusion noise. The Gaussian noise was added only to the positions of the points. Since the model also uses the normals of the points, we can distinguish two cases: when the original normals are used for each point and when the normals are recomputed in the same manner as they would be calculated on real data obtained from a camera.

6.2.2. Synthetic Dataset Trained with Noise

Using the unmodified dataset for testing showed us how a model behaves in an isolated scenario. Real-life data often includes all types of noise. In order to simulate real-life scenarios, we added Gaussian noise to Shapenet and rotated the point clouds by degrees in the range [−180, 180].

## 7. Discussion
### 7.1. Classification
7.1.1. Synthetic Dataset without Noise

Figure 13a shows that the original RGCNN network had a much better accuracy than the rest when noise is small. This is because the large enough position noise causes massive changes in the distance graph and point coordinates. For the position noise tests, increasing the standard deviation of the noise lead to a sharp decrease of the performances for all networks. A visual representation of the distortions caused by position noise is shown in Figure 5.

For the rotation noise datasets analyzed in Figure 13b, we see the effect of rotation-invariant networks. RGCNN had better accuracy for random rotations in the range [−15, 15]

degrees but its performance degraded fast with the increasing range of random rotations, whereas all the rest had stable performance regardless of the rotations applied on the point clouds. This happens because large enough rotations cause significant changes in the coordinates that are used as input for the first convolution. Rotation-invariant networks have similar input coordinates irrespective of the rotation.

Finally, for occlusion noise, we see that the original RGCNN maintains high performance, and, apart from the network using the Gram matrix, all networks maintain constant accuracy despite the larger radius of the occlusion. The occlusion changes the distribution of the points in the point cloud, which leads to a significantly different Gram matrix.

### 7.1.2. Synthetic Dataset with Noise

An interesting observation appears when looking at the rotation noise results for the models trained with position noise. RGCNN again dropped in accuracy for rotations larger than 10 degrees, whereas all the other networks remained stable. While the graph computed from the distances between the points remained the same, the coordinates of the points used in the first convolution changed, essentially creating 'new' point clouds that the network was not trained on. This appears in both Figures 13b and 14b. The other methods either place the input point cloud in a canonical position, on which the model was trained, or create new rotation-invariant features instead of the position (Gram matrix). This means that the rotation-invariant networks are now robust to large noise, while maintaining the same performance regardless of rotation noise.

Networks trained with rotation noise improved their accuracy results. RGCNN now has a sharp drop in performance at 20 degrees, whereas all the rest have higher performance than the networks trained without noise.

Lastly, for networks trained with occlusion noise, the performance is much more stable. The Gram matrix rotation-invariant network still has much better results on the dataset without occlusion, hence this method is vulnerable to occlusion noise.

### 7.2. Part segmentation

### 7.2.1. Synthetic Dataset without Noise

As can be seen in Figure 15a, all models behave similarly under added Gaussian noise if we use the original normals from Shapenet. In all of the cases, however, the mIoU accuracy decreases significantly if the normals are recomputed in the same manner that they would be in a real-life scenario (Figure 15b). Similar to the Classification case, in Figure 15d, it is shown that the original model does not hold well for large rotations, but when we rotate the point clouds according to their eigenvectors, we obtain stable performance. Occlusion noise impacts the bounding box rotations model the most but, as can be seen in Figure 15c, no considered model is invariant to this type of noise. Still, for smaller obstructions of the point cloud, the model achieves good results.

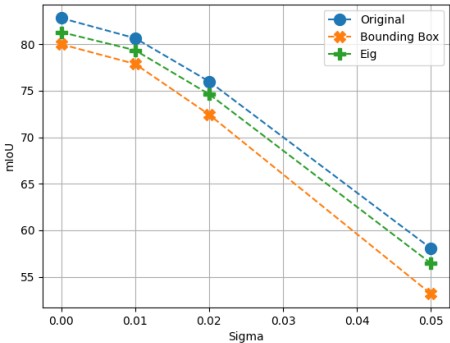
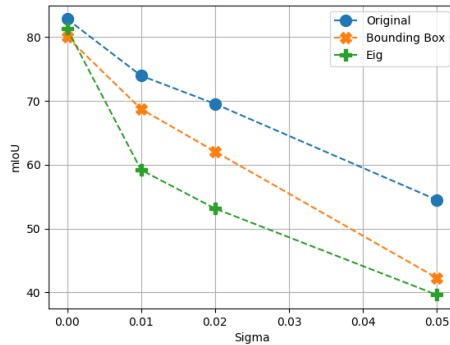

(**a**) Gaussian noise with original normals

(**b**) Gaussian noise with recomputed normals

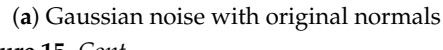

**Figure 15.** *Cont.*

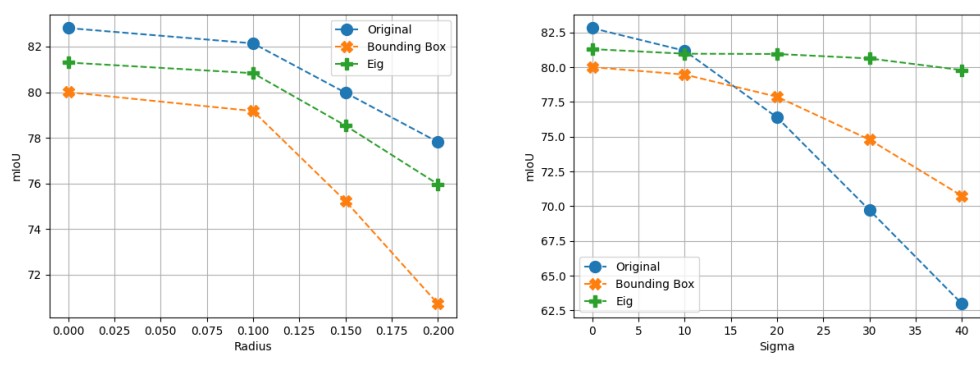

(**c**) Occlusion noise        (**d**) Random rotation noise

**Figure 15.** RGCNN with and without rotation invariance tested on noisy datasets.

### 7.2.2. Synthetic Dataset with Noise

Figure 16 shows a comparison between preprocessing the input point cloud using bounding box and eigen under different types of noise. For the first model, no other rotation normalization is applied. For the second model, the point cloud's rotation is first normalized using the bounding box method, while in the last case, we normalize the rotation using the eigenvector matrix. The first row shows the raw point cloud, without any additional noise. In the second row, a random rotation is first applied to the point cloud. The clean RGCNN model does not perform well in this case, while the other two models show superior results. In the third row, Gaussian noise is added to the position of the points. Interestingly, the eigenvector matrix normalization model completely miss-labels the plane's engines for this particular point cloud. The last row shows the results after the point cloud is occluded, missing most of the right wing.

The results shown in Figure 17a show that the previously discussed models perform very poorly when two types of noise are present. One way to improve the accuracy of mIoU is to train the models on datasets augmented with both Gaussian noise of various levels and random rotations in the range [−180, 180]. Since either of the orientation normalization procedures place the point cloud in one of nine possible orientations by first randomly rotating the point clouds before the normalization takes place, we essentially increase the number of point clouds in each of the nine canonical positions. The improvement is clearly visible in Figure 17b.

### 7.3. Model Profiling

We tested the network architectures and preprocessing methods in the same environment on an Nvidia AGX Xavier 32 GB embedded platform. All network were tested on point clouds with 512, 1024, and 2048 points and the results can be seen in Table 2. The model inference was conducted on the GPU, while the preprocessing part was done on the CPU, with the exception of the Gram matrix, which was completed on GPU. The Gram preprocessing converts an input point cloud of dimension $N \times 3$ into a set of features of dimension $N \times N$, which increases the model size and inference time. For networks using multiple views, the forward time is increased due to the extra computation from each of the four views. The preprocessing for multi-views is the same as for single views, but the convolutions are made on different views of each point cloud. In addition to the Gram matrix, all networks remain stable in size and number of parameters, as can be seen in Table 3.

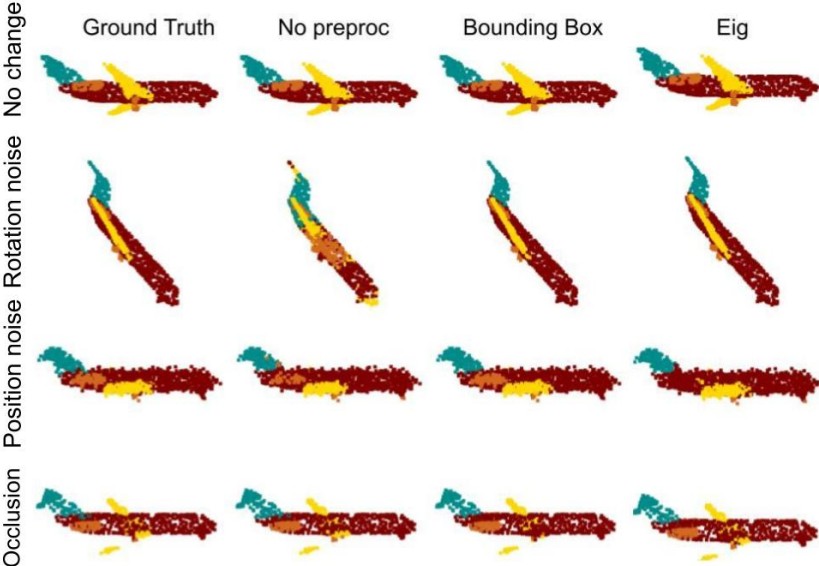

**Figure 16.** Visual comparison of part segmentation outputs for models trained with different pre-processing techniques under different types of noise. The first line shows the raw point cloud. The point cloud in the second line is first rotated by random values on each axis and then fed into the models. As can be expected, the raw RGCNN model performs the worst, while the models trained with bounding box and eigenvector matrix rotation normalization perform similarly. In the third row, we added Gaussian noise to the position of the point cloud's points. For this particular point cloud, the model trained with eigenvector matrix rotation normalization seems to mislabel the whole engine of the airplane. The last row show the results of the models applied on an occluded point cloud. All of the models perform similarly in this case.

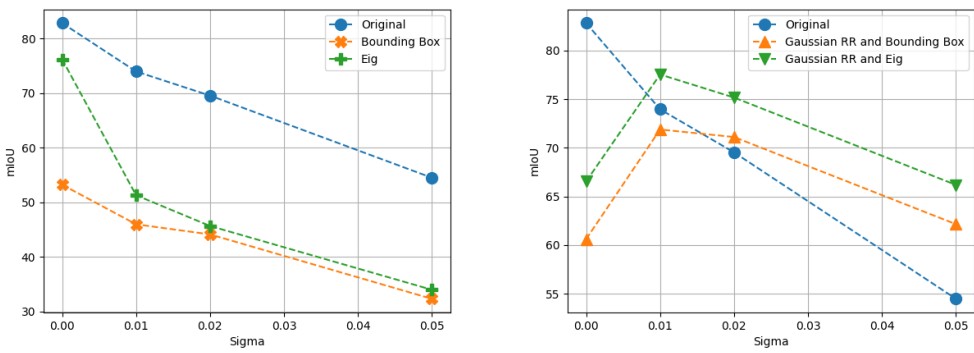

(**a**) Networks trained without Gaussian noise      (**b**) Networks trained with Gaussian noise

**Figure 17.** RGCNN with and without rotation invariance tested on a dataset simulating real-life scenarios.

**Table 2.** All classification tests have been conducted on a Nvidia AGX Xavier 32 GB embedded platform. The inference of the models was conducted on the GPU, while the preprocessing step used the CPU, with the exception of the Gram method, for which we used the GPU. As can be seen, the unmodified RGCNN network is the fastest for small point clouds, while Pointnet maintains almost constant speed for a higher number of points. Furthermore, for 512 points in the point cloud, the bounding-box preprocessing method is slightly faster than the eigenvector matrix method, but becomes slower than the eigenvector matrix method when we increase the number of points. Gram matrix is considerably slower than the other two methods in the preprocessing stage.

| Method | Nr. Points | Preproc (ms) | Inference (ms) | Total (ms) |
|---|---|---|---|---|
| Pointnet | 512 | - | 6.28 | 6.28 |
|  | 1024 | - | 6.88 | 6.88 |
|  | 2048 | - | 8.46 | 8.46 |
| RGCNN | 512 | - | 4.91 | 4.91 |
|  | 1024 | - | 12.09 | 12.09 |
|  | 2048 | - | 57.39 | 57.39 |
| RGCNN BB | 512 | 0.3 | 4.91 | 5.21 |
|  | 1024 | 0.5 | 12.09 | 12.59 |
|  | 2048 | 0.76 | 57.39 | 58.15 |
| RGCNN PCA | 512 | 0.41 | 4.91 | 5.32 |
|  | 1024 | 0.4 | 12.09 | 12.49 |
|  | 2048 | 0.47 | 57.39 | 57.86 |
| RGCNN Gram (GPU) | 512 | 0.85 | 6.07 | 6.92 |
|  | 1024 | 2.16 | 23.75 | 25.91 |
|  | 2048 | 10.71 | 143.22 | 153.93 |
| BB multi view | 512 | 0.3 | 12.45 | 12.75 |
|  | 1024 | 0.5 | 30.8 | 31.3 |
|  | 2048 | 0.76 | 103.49 | 104.25 |
| PCA multi view | 512 | 0.41 | 12.45 | 12.86 |
|  | 1024 | 0.4 | 30.8 | 31.2 |
|  | 2048 | 0.47 | 103.49 | 103.96 |

**Table 3.** Model size and number of parameters for Pointnet, DGCNN, RGCNN, and RGCNN with Gram matrix for the classification task. We consider the case where the input point cloud has 512 points. DGCNN has the smallest number of parameters. The Gram matrix RGCNN has a higher number of parameters and a larger model size than normal RGCNN because Gram preprocessing increases the size of the first convolution weight matrix from $N \times 3$ to $N \times N$.

| Network | Model Size (MB) | Number of Parameters |
|---|---|---|
| Pointnet | 14 | 3.478.796 |
| DGCNN | **7** | **1.809.576** |
| RGCNN | 16 | 4148680 |
| RGCNN Gram | 18 | 4539592 |

## 8. Conclusions

We used a new rotation normalization algorithm based on oriented bounding boxes to achieve rotation invariant for classification and semantic segmentation. We created an annotation tool for part segmentation on real camera data by using bounding-box rotation normalization. Classification and part segmentation networks were successfully trained on real camera data and deployed on an embedded device. Tests on synthetic datasets have shown that the networks are invariant to random rotations. By augmenting the training set with noisy data, the network performance was improved for Gaussian noise.

The network offers new estimations about the occupied volume and orientation of the point cloud which other rotation invariant methods do not provide. This is relevant for a number of tasks, including segmentation and classification pipelines. We aim to extend the network capability to multi-object classification and multi-object part segmentation for embedded devices. With a run-time efficient and robust method against noise, the results of the proposed algorithm are comparable with the ones from the main literature.

**Author Contributions:** Conceptualization, A.P. and L.T.; funding acquisition, L.T.; investigation, A.P.; resources, A.P. and V.D.; software, A.P. and V.D.; supervision, L.T.; validation, A.P.; visualization, A.P. and V.D.; writing—original draft, A.P. and V.D.; writing review and editing, L.T. All authors have read and agreed to the published version of the manuscript.

**Funding:** This research was funded by Unitatea Executiva Pentru Finantarea Invatamantului Superior Si A Cercetarii Stiintifice Universitare, grant number PN-III-P2-2.1-PED-2021-3120.

**Acknowledgments:** The authors are thankful for the support of Analog Devices Romania, for the equipment list (cameras, embedded devices) as well as the GPU cards from Nvidia offered as support to this work.

**Conflicts of Interest:** The authors declare no conflict of interest.

## Abbreviations

The following abbreviations are used in this manuscript:

| | |
|---|---|
| CNN | Convolutional Neural Network |
| RGCNN | Regularized Graph Convolutional Neural Network |
| DGCNN | Dynamic Graph Convolutional Neural Network |
| RGB-D | Red–Green–Blue-Depth |
| mIoU | Mean Intersection Over Union |
| kNN | K-Nearest Neighbor |

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
