# Peer review of "Rotation Invariant Graph Neural Network for 3D Point Clouds"

_remotesensing, doi:10.3390/rs15051437_

Round 1
Reviewer 1 Report
PCA boxing is known to the point cloud community, also matching the eigenvectors of graph Laplacian (also rotation invariant) and often it is applied to PCs without a further ado. This paper provides an interesting comparison between rotation invariance built in an NN and pre-processed orientation normalizations.
Especially computational costs would have been in interest to the research community, since methods vary from heavy global formulations to fast localized filtering. There is one table for computation times for one method (RGCNN in some of its forms) in different computing environments, but since methods behave so differently under pointwise noise/occlusion/position noise/rotation noise (each case bringing some success and some failure scorings to different methods), it would be beneficial to get rough estimates on the time complexity of the following:
- eigen values ("Eig". in Fig 12)
- RGCNN in its various forms (only prediction, not teaching)
- multiview
p. 9: adnotate --> annotate
So, I recommend to add some time complexity summary. Exclude Nano environment, if a method simply does not fit in Nano...
The target are individual compact objects, which traditionally have been oriented by PCA before the PC analysis. The main question seems to be comparing the quality of various approaches, not so much the computational efficiency, although computation times of one method on 3 different systems has been given.
The analysis of efficiency of different approaches under rotational, translational and Gaussian noise are interesting, since these are reflected with a modification of the given method (RGCNN) and various preprocessing phases (Gram, PCA, eigenvectors). Very often a new method is presented without a proper consideration of all possible preprocessing alternatives, and since a parallel implementations are common in industry, this paper is a first bird in that sense.
It points out that a simple preprocessing can sometimes improve The main deliverable is the orientation information and implementability on an embedded device.
Since the target is embedded environments, the computational speeds of alternatives (Fig. 9) should be somehow described. E.g. how costly ids to genrate an eigen vector matrix (on one environemnt, I doubt it can be done on Jetson Nano?) . I repeat, computational complexity comparison is important to see which alternatives are embedded or edge -computing friendly.
Conclusions are satisfying, I dont know ModelNet40 well, but was there any problems with relatively "roundy" objects (which have similar fst 3 eigenvalues), since such objects could produce errors in repeated orientation detection?
References are ok.
No additional comments. A rather good paper, maybe a bit too busily written.
Finally, I am sorry to provide my input too late, I was mislead by an internal status list of reviews.
Author Response
Please find the attached response to review.

Reviewer 2 Report
1, there are currently many graph convolutional networks for point cloud processing, why did the author take RGCNN?
2, in fact, the method proposed by the author can be regarded as a treatment method for obtaining the rotational invariance of the point cloud. Why is there no comparison with other treatment methods to prove the effectiveness and superiority of the method proposed in this article?
3. In terms of article organization structure, it is recommended to put the two subsections 3.1 and 3.2 in the background part, rather than the method part.
Author Response
Please find the attached response to the review.

Reviewer 3 Report
A novel rotation normalization technique for point cloud processing using an oriented bounding box is proposed in this paper,and was used to create a point cloud annotation tool for part segmentation on real camera data.
1. Oriented boundary box is an important part of the research method in this paper. The author mentioned it in line 123, but how to calculate the boundary box is not mentioned. The calculation process of oriented boundary box should be given.
2. I think in 6.1. Comparison between server, laptop and embedded is unnecessary. The better the performance of hardware equipment, the higher the efficiency of calculation. If author would like to make a comparison of the results, I suggest that author needs to make some comparisons with the results of rotation invariant for classification and semantic segmentation methods used in this paper to highlight the advantages of this research method.
3. In the article, more line charts (e.g. Figure 10, Figure 11) are used to represent the accuracy of classification and semantic segmentation, however, charts of point cloud test data and ground truth comparison should be added to more intuitionistic performance. Please refer to the article Regularized Graph CNN for Point Cloud Segmentation.
4. Please explain why RGCNN was chosen when the Mean Class Accuracy and the Overall Accuracy of DGCNN were better than RGCNN. Please refer to the article Regularized Graph CNN for Point Cloud Segmentation.
5. To evaluate the performance of a classification or segmentation network, in addition to Accuracy and mIoU, there are also Model Size, loss changes in model training, FLOPs and so on. I hope the author could reflect these indicators in the article.
6. I hope the author could explain why the accuracy of RGCNN changes sharply in Figure 10 (b), Figure 11 (b) and Figure 11 (c), while other methods do not.
7. The point cloud information used in the article does not contain RGB information. If RGB information is added, the accuracy of the test results will be higher in theory. I hope the author could explain why RGB information is not used.
8. I hope the author could explain the rationality of the three types of noise settings. In this paper, the three types of noise are analyzed one by one. The author should consider the combination of the three types of noise, such as Occlusion noise and Gaussian noise.
9. In an algorithm flow, we are concerned about the input and output of the algorithm, so I hope the author could explain what fin and fout in Figure 2. Spectral graph convolution represents respectively.
10. The mathematical expression written by the author in the article is not standardized (e.g. 3.2. Background for graph neural networks, 3.3. Spectral graph convolution neural network). The mathematical expression should be started a new line and be written in the order of serial number.
Author Response

(The authors gave the same response as above.)

Round 2
Reviewer 1 Report
If Table 1 includes actual results, it should be moved to results. Otherwise (if Table 1 illustrates profiling of the existing methods) I recommend publication as-it-is.
Author Response
Thank you for the suggestion, we implemented it in the final version of our work!
Reviewer 2 Report
authors have addressed all my concerns.
Author Response
Thank you for the positive feedback!